# Reproductive Biology of Dry Grassland Specialist *Ranunculus illyricus* L. and Its Implications for Conservation

**DOI:** 10.3390/biology11060873

**Published:** 2022-06-06

**Authors:** Dawid Kocot, Ewa Sitek, Barbara Nowak, Anna Kołton, Krystyna Towpasz

**Affiliations:** 1Department of Botany, Physiology and Plant Protection, Faculty of Biotechnology and Horticulture, University of Agriculture in Krakow, 29 Listopada 54, 31-425 Krakow, Poland; dawid.kocot@urk.edu.pl (D.K.); barbara.nowak@urk.edu.pl (B.N.); anna.kolton@urk.edu.pl (A.K.); 2Department of Ornamental Plants and Garden Art, Faculty of Biotechnology and Horticulture, University of Agriculture in Krakow, 29 Listopada 54, 31-425 Krakow, Poland; 3Department of Plant Ecology, Institute of Botany, Faculty of Biology, Jagiellonian University, Gronostajowa 3, 30-387 Krakow, Poland; krystyna.towpasz@uj.edu.pl

**Keywords:** tubers, progeny plants, *Illyrian buttercup*, clonal plants

## Abstract

**Simple Summary:**

The Ranunculus illyricus—Illyrian buttercup—is threatened with extinction in many countries and measures should be taken to protect it. In order to increase the effectiveness of such measures, it is necessary to know the methods of propagation and to evaluate their efficiency. *R. illyricus* reproduces generatively by seed and vegetatively by clusters of progeny tubers. The method and potential of vegetative propagation are described here for the first time and compared with the potential and actual effectiveness of generative propagation. Both the generative and vegetative propagation methods should be used to strengthen existing populations and create replacements.

**Abstract:**

*Ranunculus illyricus*, a component of xerothermic grasslands, is a declining species and deserves active conservation treatments in many countries preceded by studies on the biology of its reproduction. So far, our knowledge of *R. illyricus*, a species with two modes of reproduction, has been fragmentary. The purpose of the studies presented here was to describe the annual development cycle of *R. illyricus* with particular emphasis on the production of underground tuber clusters that serve as vegetative propagation. Based on three-year-long observations in an ex situ collection, the efficiency of vegetative propagation was estimated and compared with the efficiency of generative propagation. It was found that in 3 years the best clones could produce up to 57 progeny clusters followed by flowering specimens in the first season. Meanwhile, the high potential for generative reproduction was suppressed by many limitations including fruit setting, the germination capacity of seeds, seedling survival rate, and additionally, the first flowering plant was observed only in the third year. It seems that the efficiency of vegetative propagation of this species can be higher than the efficiency of generative propagation. Moreover, vegets bloomed in the first year after emergence, whereas the first plant of generative origin was observed to bloom only after 3 years. A large proportion of individuals of vegetative origin can negatively affect the genetic diversity of the population but their survival rate against competing plants is higher. To enhance the existing populations or to create new ones, it would be best to use plants derived from clonal propagation of genets carried out in ex situ conditions.

## 1. Introduction

Anthropogenic pressure in the environment leads to the loss and fragmentation of natural habitats, invasion of alien species, overexploitation of resources, environmental pollution, and climate change. As a result, more and more species are becoming extinct. According to various studies, it is estimated that about 8% [1] up to one-third of plant species are at risk of extinction, including most of those that have not yet been described given their limited ranges and local rarity [2,3]. In many cases, it is necessary to implement active conservation programs, and their effectiveness depends on the extent to which the biology of the species and threats in the environment are recognised [4]. Reproductive biology directly affects wild populations because reproductive success and certain life cycle traits (longevity, offspring recruitment, and survival) determine population survival and growth, whereas the life form and mode of reproduction affect the level of genetic diversity and distribution within and among populations [5]. 

Currently, natural and semi-natural grasslands are highly threatened plant communities as a result of conversion to cropland, afforestation, spontaneous succession, and urbanisation. It is estimated that at least 50% of all grasslands in Central and Eastern Europe have disappeared in the last 200 years. These communities are extraordinary refuges of biodiversity. In Europe, 29% of all bird species are associated with grassland habitats, calcareous grasslands and steppes are home to 63% of butterfly species, and among Europe’s >6000 endemic vascular plant species, grassland species make up the second-largest group (18.1%) [6,7]. 

One of the species inhabiting xerothermic grasslands, steppes, and dry sunny slopes is *Illyrian buttercup* (*Ranunculus illyricus*), a clonal herb in the family Ranunculaceae, section Ranunculastrum, subgenus Ranunculus (*Ranunculus*) [8]. It is classified as a dry grassland specialist and referred to as an indicator species of “flowery, meadow steppes” [9,10,11]. It also occurs in synanthropic habitats, such as burial mounds [12] or old graveyards surveyed in the Pannonian ecoregion [13,14]. 

The main range of occurrence covers most of the countries of the Balkan Peninsula, central Italy, Romania, Ukraine, the European part of Russia (in the south), and Turkey. Scattered populations occur in many European countries, such as Sweden, Germany, the Czech Republic, Slovakia, Hungary, and Slovenia, where it is most often considered an endangered species [15,16]. In Poland, it is a rare and critically endangered species subject to legal protection and currently occurs only in two locations [17,18]. The resources of this species, closely linked to xerothermic grasslands, are decreasing throughout its whole range. 

The *Illyrian buttercup* is easy to recognize due to strong, grey-white hairs on the aboveground parts, also on its three-petaled leaves. It is a perennial classified as a geophyte with a relatively short growing season from April to June, and a flowering period typically between May and June; yet, soon after fruiting, the plant dries up. It is a monoecious, bisexual, and insect-pollinated plant, which reproduces by seeds or vegetatively using underground stolons at the base of the shoot [16,19]. Flowers typical of the genus *Ranunculus*, with the calyx sepals characteristically bent downwards, are self-incompatible [19]. The fruits of this species are one-seeded achenes, whose morphology and anatomical details were described by Mourad et al. [20] and Gherghişan [21]. 

The *Ranunculus* genus comprises species rich in alkaloids with essential oils containing a high percentage of fatty acid (mainly hexadecanoic acid), phytol, and hydrocarbons [22,23,24]. The herb of *R. illyricus* is used for medicinal baths because it contains coumarins (umbelliferone) with antibacterial properties [25]. 

Although *R. illyricus* is known to develop stolons and storage roots, there is no information available on the developmental biology of its underground parts and the recruitment mode of vegetative progeny. As an endangered species in most of its range and as a potentially useful plant (medicinal and ornamental), *R. illyricus* deserves a more detailed understanding of its reproductive biology. The objectives of this study are (A) to describe the biology of the species, and (B) to assess and compare its ability to reproduce generatively and vegetatively in order to verify the hypothesis that the two reproduction modes of this species are equally effective. Understanding the plant’s reproduction system can help conservation and management strategies because plant populations may be greatly impacted by limitations related to generative or vegetative reproduction. Reproductive output is defined as the number of individuals produced in the following growing seasons, i.e., seedlings in generative reproduction (genets) and clusters of progeny tubers (PC) in vegetative reproduction (ramets).

## 2. Materials and Methods

### 2.1. Plant Material and Growth Conditions

The research was carried out on an ex situ population of *Illyrian buttercup* (*R. illyricus*) grown in a collection at the Faculty of Biotechnology and Horticulture of the University of Agriculture in Kraków. The cultivated plants represented the population coming from Miernów—one of the Polish natural populations [16,17]—and the plant material was received from the Botanic Garden of the Jagiellonian University in Kraków. 

Individual tagged plants were grown in pots of a 7 cm diameter containing peat substrate and deacidified peat (1:1). That was the standard soil used throughout the experiment. Plants were cultivated in open-field conditions, naturally covered by snow during winter. The pots with the plants were watered and over the years of observations, the data describing the thermal conditions in the collection were recorded. The experiment was established in the autumn of 2016 and observations of aboveground and underground parts were conducted for three consecutive seasons (in 2017 and 2018 for all clones, whereas in 2019 for randomly selected clones) (Figure 1). The observations allowed describing the annual development cycle under ex situ conditions.

### 2.2. Vegetative Reproduction

In autumn 2016, 39 tuber clusters were planted. Each year, during the dormancy period (September/October), the plants were taken out of the pots and the development of the underground parts was evaluated: the number of progeny tuber clusters (PC), the fresh matter (FM) of the mother (MC), and progeny tuber clusters (PC), as well as the number of tubers in the cluster. Each cluster was then replanted separately in new pots and labelled so that the clones derived from a single parent plant could be tracked throughout their development. The data collected in this way allowed us to not only describe the vegetative development of individual clones but also to compare the underground organs (tuber clusters) produced during three consecutive years in each age group of clusters: new, one-year, and two-year clusters of tubers. The clusters labelled “new” were those that were produced as new progeny in a given season, e.g., the clusters planted in the autumn of 2016 had produced entirely “new” clusters by the autumn of 2017. Those replanted in the autumn of 2017, in turn, were “one-year” clusters in 2018. Throughout the manuscript, a cluster of tubers represents one individual (plant). What is meant by vegetative offspring is a progeny cluster of tubers (PC)—ramets—produced by the mother cluster of tubers (MC).

### 2.3. Generative Reproduction

During flowering, the proportion of flowering individuals in the population, the height of flowering stems, and the number of flowers per plant were assessed (2018, 2019); in the tagged flowers (n^2019^ = 20), the stamens and pistils (n^2017^= 25; n^2018^ = 24; n^2019^ = 72 flowers) were counted followed by the fruit set. The efficiency of seed formation was calculated as the ratio of the number of mature achenes to the number of all pistils, expressed as a percentage.

#### 2.3.1. Pollen Quality (Production and Viability)

The anthers were isolated from the breaking buds of the non-tagged plants (Figure 2h) to evaluate the quality of the pollen. The viability of the pollen was assessed on the day of harvest by the indirect staining Alexander method [26], in which red-stained pollen is considered viable and green is considered unviable. Data were collected during three growing seasons. Assessments were made in three replicates, each prepared as a pollen mixture from 10 flowers. The viability of 300 pollen grains from each replicate was assessed using a Zeiss Axio Imager M2 microscope (Carl Zeiss, Jena, Germany). Photographs were taken using an EOS 450D Digital Camera (CANON, Tokyo, Japan).

Pollen production was estimated for the stamens that were sampled before anther dehiscence (Figure 2h). A mixture of 10 stamens from 10 flowers was placed in a 1.5 mL Eppendorf tube and dried at room temperature. After dehiscence, 1.0 mL of distilled water was added and the content was vortexed immediately prior to counting pollen grains using a Bürker haemocytometer [27]. The number of pollen grains in one anther was calculated according to the following formula:
A = (X × Ve)/(Vc × n)
A = (X × 1000 μL)/(0.1 μL × 10)where:
A—number of pollen grains per anther,X—average number of pollen grains per counting field,Ve—total volume of the pollen grain solution in a 1 mL (1000 μL) Eppendorf tube,Vc—volume of the counting field of 0.1 mm^3^ (0.1 μL),n—number of stamens in the Eppendorf tube.

#### 2.3.2. Seed Viability and Ability to Germinate

The viability was assessed for 30 seeds using the tetrazoline method [28]. The achenes were soaked in water for 24 h, then the pericarp was removed and seeds were soaked in tetrazolium for another 24 h. The seeds whose endosperm and embryo were completely red were considered to be alive, whereas those that were partly red or white were treated as dead. 

Ripe one-seeded achenes were tested for their ability to germinate using the blotter test in Petri dishes in conditions of a 16/8-hr day/night photoperiod [29]. The influence of the factors interrupting the seed dormancy was evaluated. In 2017, the effects of germination temperature (10 °C or 20 °C) and warm stratification were evaluated immediately after harvest (soaking in a 50 °C bath for 2 min prior to sowing). The impact of low-temperature stratification (4 °C for a period of four weeks) and pre-sowing conditioning with gibberellic acid (1.0 × 10^−3^ mM, for 24 h) was evaluated in 2018. The results were recorded after four weeks. The experiment was repeated four times for each factor (combination) using four Petri dishes with five seeds for each combination.

#### 2.3.3. Development and Survival of Seed-Derived Plants under Ex Situ Conditions

A month after sowing (18 July 2018), 25 seedlings with cotyledons obtained from the blotter test were planted individually into pots in the same soil as the mother plants. They were cultivated for 4 weeks in a Sanyo vegetative chamber (Sanyo-Onoda, Japan), under a 16/8-hr day/night photoperiod and photon flux density of 45 µmol m^−2^ s ^−1^, a temperature of 24 ± 2 °C, and humidity of approximately 60%. The plants were then moved to an unheated greenhouse and in spring, after overwintering, transferred to a field collection. The survival rate and development of aboveground and underground organs of the seed-derived plants were observed in subsequent growing seasons (2018–2021) and the number of clusters and tubers per cluster were recorded. 

### 2.4. Statistical Analysis

All statistical analyses were performed with STATISTICA v. 13.3. The normality of data in groups was tested with the Shapiro–Wilk test. The homogeneity of variance in the groups was tested by employing the Levene test. When comparing the groups with a normal distribution and homogeneity of variance, parametric tests (ANOVA and Tukey or Student *t*-test) were used. However, when comparing the groups characterised by the lack of a normal distribution or the lack of homogeneity of variance, nonparametric tests (Kruskal–Wallis and Dunn’s test or -Mann–Whitney U test) were applied. A correlation analysis was also performed where the Pearson correlation coefficient r was determined. Details of the tests used are provided in the captions of tables or figures. The significance level was α = 0.05.

## 3. Results

### 3.1. Annual Development Cycle

Under experimental conditions, *R. illyricus* develops a system of fibrous roots underground, some of which accumulate storage materials (starch) and develop into tuberous roots (Figure 2b–e). The unmodified roots are annual, whereas the storage roots form perennial tuber clusters with a perennating bud in the central part. In addition, the stolons, which are the organ of vegetative reproduction of this species, can form at the base of the developing bud in autumn (Figure 3).

At the end of June, the whole aboveground part, the unmodified roots, and the underground stolons connecting the clusters of mother tubers with the progeny die. The plant takes the form of a cluster of tubers with a dormant perennating bud on top (Figure 2a). Dormancy occurs in summer until the beginning of autumn (July–September) and lasts for at least three months. At the end of September, new roots emerge at the base of the regenerating bud, followed by underground shoots. The dormant bud meristem also becomes active and starts to grow (Figure 2b,c). One month later (in October) the roots reach about 10 cm. At this time, the stolons are shorter than the roots (up to 5 cm), and are thicker and more rigid (Figure 2c,d). In November, the tuber clusters have a developed abundant root system and several-centimetre-long shoots—greenless and hidden under the soil surface (Figure 2d).

In some seasons, the shoots develop leaves above the soil surface and thus overwinter (Figure 2e). The further development of the aboveground parts proceeds in parallel with the further development of the stolons and the formation of progeny clusters. At the end of April, the aboveground vegetative shoots are developed and the stolons end in clusters of progeny tubers underground. The stolons reach a length of 15–20 cm and are divided into 4–5 internodes. At the nodes, single, reduced leaves are visible in the form of scales. Clusters of progeny tubers are formed at the top of the stolon (always one cluster on one stolon). The progeny cluster has only one regenerating bud in the central part, opposite the stolon (Figure 3). The growth of inflorescence shoots occurs in May, whereas the beginning of flowering falls in the third week of May (Figure 2f). The flowering of the population lasts about 3 weeks (to the beginning of June), and the flowering of a single flower, from the opening of the bud to the falling of the perianth and stamens, takes 6–8 days (Figure 2g–j). After ripening and drying of the fruit (Figure 2k), the whole aboveground part dies and the plant starts the summer dormancy period again. 

### 3.2. Vegetative Reproduction

Labelled tuber clusters and their annual monitoring allowed us to track the life history of individual clones and to estimate their ability to reproduce vegetatively in subsequent years. Of the 39 tuber clusters planted in 2016, 90% had survived by 2017, 70% by 2018, and 64% by 2019. This shows that *R. illyricus* ramets can survive at least 3 years under ex situ conditions. 

The potential for vegetative propagation was detailed using the example of the clone designated “11” over three years (Figure 4). Each season, an average of 3.3 progeny clusters were produced per plant, with a maximum of 5. During this time, a total of 57 progeny clusters (ramets) were produced from a single mother cluster (clone 11). We observed that each ramet could produce progeny clusters for at least two years (Figure 4).

However, the clones differed in vegetative potential. Some of the plants died off without forming progeny clusters. In general, in the plants that had reproduced vegetatively, an average of 8 (between 3 and 19, depending on the clone) progeny clusters were formed from a single cluster after 2 years, and after 3 years 13–51.

It was not only the clone that affected the number of PC developed but also the age of the cluster. The age of the cluster also determined its FM, the number of tubers in the cluster, and the number of flowers per plant. One-year-old clusters had the highest number of tubers, which was much lower in the two-year-old clusters as was the FM of the cluster (Table 1). Younger tuber clusters also produced more flowers, although not all flowers in both age groups set fruit (Table 2).

To determine whether the size of planted clusters can determine the mode of reproduction in the next season, the plants were evaluated in two age groups in 2018. The plants indicated as ”new” in 2017 (*n* = 64) are “one-year-old” in 2018, whereas the “one-year-old” plants in 2017 (*n* = 31) are “two-year-old” in 2018 (Table 3). It turned out that most individuals reproduced both vegetatively and generatively in 2018 (VG in Table 3). In the group ”one-year-old” in 2018, 63% reproduced this way, whereas in the group “two-year-old” in 2018, it was 49%. However, there were some that reproduced only vegetatively (36 and 35%, respectively), or only generatively (3 and 16%, respectively). 

It turned out that the FM of one-year tuber clusters did not co-vary the mode of reproduction in the following year. However, the FM of the younger clusters—“new” clusters—did. In that case, clusters with a higher FM produced individuals that reproduced only generatively (G), and clusters with a lower FM produced individuals that reproduced only vegetatively (V)—Table 3. Moreover, next year in this age group, the tuber FM of the VG individuals was significantly higher than that of the V individuals, which may be due to differences in the FM of the clusters planted a year earlier. However, the VG plants also produced more clusters of tubers (2.7) than the plants reproducing only clonally (1.6). It is puzzling why flowering individuals produced more clusters of higher FM. It could be expected that the allocation of resources to the organs of generative reproduction will have a negative effect on the tuber FM. However, it seems that the additional photosynthetic area of leaves on a flowering shoot meets the needs of generative reproduction as well as the accumulation of storage materials. Plants that reproduce only vegetatively develop only a rosette of leaves. 

The height of the inflorescence shoot, the number of flowers, and the percentage of fruit-bearing flowers were the same regardless of the mode of reproduction (G or VG) in both age groups (Table 3).

Temperatures in the months when the development of the aboveground part took place (March, April, and May) affected the FM of the clusters. Higher temperatures in March positively influenced their FM, but higher temperatures in April and May decreased the tuber FM (Table 4, Figure 5). The data suggest that March can be the main month for tuber formation. Assuming optimum plant watering, the negative effects of higher April and May temperatures can be explained by the intensification of developmental processes other than the accumulation of storage materials, for example, faster shoot growth or more numerous flower buds.

### 3.3. Generative Reproduction

#### 3.3.1. Flowering and Fruit Setting

During the 2018 growing season, 57.6% of the individuals flowered, whereas in 2019, 85.0% of the individuals flowered; the non-flowering individuals developed only a rosette of leaves. The flowering plants had a single stem with one to four flowers, yet some of the flowers did not set fruit—Table 3. On average, the number of stamens was 66, whereas the number of pistils was 147 (138–156), without seasonal variations. The viability of pollen was 53.6–68.5% depending on the year (Table 5). The diameter of the alive (red) pollen grains was larger than that of the unstained, dead pollen grains (Figure 6a). The production of pollen was abundant as almost 140 thousand pollen grains were obtained from one flower (Table 5). 

The species is characterised by a high potential for generative reproduction due to the high number of developed pistils in the flower, the number of stamens, and numerous pollen grains with relatively high viability varying across seasons. On the other hand, the seed-setting efficiency, although varying from season to season, was low (the highest was 12.8%) and therefore a small number of fruits formed from numerous pistils (Table 5).

#### 3.3.2. Seed Germination

The seeds of the *Illyrian buttercup* were 100% viable (Figure 6b) but germinated with difficulty. The best germination rate was obtained when the seeds were germinated at a reduced temperature (10 °C), and also those that were subjected to cold stratification germinated relatively well (Table 6 and Table 7).

#### 3.3.3. Development of Seedlings under Ex Situ Conditions

*R. illyricus* seeds germinated between 10 to 18 days after sowing. The seeds germinated epigeically and one month after sowing most of the seedlings had cotyledons (Figure 7a,b). In the first year, plants produced aboveground a rosette of juvenile leaves (3–12 leaves) (Figure 7c,d) and a fibrous root system belowground (Figure 7e). In some cases, the formation of elongated tuberous storage roots was observed in autumn (Figure 7f). In the first year (2018), the mortality of seedlings was high; only 50% survived the first winter (Figure 7g).

In the spring of the second year (2019), the active plants produced leaves but none of the plants flowered. The plants had thickened roots underground and half of them produced stolons terminated with progeny clusters of tubers but only one per plant. 

In the third year, generative reproduction occurred for the first time but only one seed-derived plant flowered. All three-year-old seedlings produced progeny clusters with 5–15 tubers (Figure 7h). In the next growing season, the number of tubers in clusters was higher, on average 15 for three- or four-year-old clusters and 9.5 for two- and one-year-old clusters.

## 4. Discussion

This article presents *R. illyricus*, a species with a wide geographic range throughout Europe and also Asia. However, in many countries, this species, associated with xerothermic grasslands, is rare and to various degrees under threat of extinction. So far, however, little is known about its reproductive biology and the available data are only fragmentary [30,31,32].

In order to carry out effective active conservation measures, it is necessary to recognize the threats not only to individual habitats but also to the biology of plant reproduction, which will fully reflect the existing causes of population decline [4,33,34]. In Poland, this species is an extremely rare plant considered to have been extinct for several decades (EX category) [16]. All the more valuable are the two populations discovered later: one is located in a steppe reserve (the Skorocice reserve) and the other on a kurgan in an agricultural landscape (the village of Miernów) [17,18]. The second population is particularly exposed to all adverse changes observed in such a type of habitat: progressive succession, eutrophication, invasion of alien species, intensive agrotechnical treatments, and isolation [35,36]. The observations described here were made on an ex situ collection representing Polish natural resources of *R. illyricus* from the Miernów site.

*Ranunculus illyricus* is a species with potentially two modes of reproduction: vegetative and sexual. An analysis of data collected over three seasons of observations allowed us to describe the full annual vegetative cycle of *R. illyricus* with particular emphasis on the development of underground vegetative organs. The underground-forming rhizomes ending in clusters of tubers are used for vegetative reproduction and as storage resources.

The number of PC produced is a measure of the efficiency of vegetative propagation. It was found that a single cluster can live for at least three years and produce PC each year. The number of clusters produced varied across clones but also depended on the age of the MC. One-year-old clusters had the highest reproductive potential and produced on average about 2.3 progeny clusters. Two-year-old tuber clusters were already characterised by a lower reproductive capacity, which may be due to their declining storage resources: they counted fewer tubers and had lower FM and they also flowered less (Table 1, Table 2 and Table 3). It has also been shown that external factors can indirectly affect the efficiency of vegetative propagation. The number and FM of tubers were positively correlated with March temperatures. Warm spring months stimulate earlier development of the aboveground vegetative part, which accumulates the produced resources at that time in the formed underground organs. Presumably, water resources could also modify the efficiency of vegetative reproduction (and this is probably the case in natural populations); however, the ex situ collection was regularly watered and observations were conducted under optimal watering conditions.

The potential efficiency of generative reproduction depends on numerous factors related to the development of the plant, such as the number of viable pollen grains formed and the number of developed pistils in the flower, as well as the number of developed flowers, among others. In the case of *R. illyricus*, numerous pistils and a large number of pollen grains with fairly high viability were formed, but this potential was not fully exploited. Although fruit-setting efficiency varied in successive growing seasons, it was always low and did not exceed 12.8%. The low fruit set may be a consequence of the lack of effective pollinating insects, but it may also be a result of a low genetic variation of the plants in the collection. The collection contained clones that reproduced vegetatively—no individuals of generative origin appeared. Since *R. illyricus* is a self-incompatible species [19], the genetic homogeneity of the population is a factor that significantly limits the efficiency of generative reproduction. In future, it should be verified what the efficiency of the generative reproduction of this species is and what the proportion of individuals of generative versus vegetative origin would be in natural populations. In the course of our observations, other limitations to generative reproduction have also been noted. Not all developed flower buds opened (Table 3) and set fruit. It is difficult to judge whether this is the result of environmental factors or perhaps a programmed dying of lateral flower buds, which has been described for *Ranunculus bungei* as well as other species in the family Ranunculaceae [37] at early stages of inflorescence differentiation.

Further limitations to generative propagation arise from the difficulty in germinating seeds that enter dormancy. Of the dormancy-interrupting factors tested, the best results were observed with a low (10 °C) germination temperature or the interaction of stratification with GA_3_ application. Based on this, it can be concluded that favourable conditions for germination in the wild are created by a cool autumn in the year of seed shedding or by the spring of the following year. A low temperature is a factor that contributes to the breakdown of abscisic acid, which is a germination inhibitor. GA_3_ is a plant-growth regulator, an antagonist of abscisic acid, and has been repeatedly used to break seed dormancy [38,39,40,41] for *R. asiaticus.* A high temperature, which positively stimulated the germination of *R. asiaticus* seeds [42], proved to be an ineffective factor in breaking the *R. illyricus* dormancy in our studies.

Assuming the most optimistic parameters affecting the efficiency of generative reproduction in ex situ conditions: the number of seeds per flower—19, and the germination capacity of the seeds—40%, it takes seeds from more than three flowers (3.3) to obtain one flowering individual after 3 years. In turn, in the example of clone 11, after one year, on average 3.3 PC develop from one MC, and after three years up to 57. In addition, plants derived from vegetative propagation take up growth and flower already in the first year after formation. With such assumptions, it can be concluded that most individuals, also in natural populations, are of vegetative origin.

The vegetative way of reproduction ensures the survival of vegetative progeny among strong competitors (i.e., grasses). In xerothermic grassland populations, vegetative reproduction is dominant [43]. Seedling development is rare because of strong competition [44,45]. Vegetative reproduction is expensive, as evidenced by the high allocation of mass to the production of vegetative progeny but ensures almost 100% reproduction success. This situation is a classical illustration of the trade-off rules—greater investment in the progeny increases their chances of survival [46]. It also happens that the efficiency of the generative reproduction of xerothermic grassland species is very high, but their populations decline in the face of strong competition and changes in habitat use [40]. In this case, the introduction of grazing or mowing on xerothermic grasslands with the removal of green matter could positively affect the survival of seedlings on natural sites and at the same time increase the genetic diversity of the population.

Two other species with similar biology can be used as a reference point to evaluate reproductive processes: *R. asiaticus* and *Ficaria verna* (i.e., *R. ficaria*). *R. asiaticus* occurs naturally on the Mediterranean coast and is widely cultivated as an ornamental plant [47]. *F. verna*, on the other hand, is a component of the spring undergrowth of deciduous forests native to Europe and Asia, whereas in North America it is an eradicated invasive species [48]. All three species can be defined as perennial geophytes with monocarpic aboveground stems adapted to seasonal climatic changes. *F. verna* was also found to exhibit low efficiency of generative reproduction [32], which does not prevent it from occurring in large numbers thanks to its efficient vegetative reproduction. It is known that in undisturbed communities this mode of reproduction dominates and effectively ensures the survival of plants, e.g., the perennial undergrowth of deciduous forests [49,50] or the steppe perennial in the temperate zone [43].

The development of the underground organs of *R. illyricus*—tuberous roots and stolons—is somewhat reminiscent of *R. asiaticus*, which also undergoes a period of dormancy during hot and dry summer months. During wet and cool months, the plant goes through a generative phase and develops stolons and tuberous roots [31,47,51]. However, the vegetative buds of this species are located in the external leaves of the rosette and give rise to the growing point of the tuberous roots. The tuberous roots can be divided but the annual multiplication rate is only 2–5 [47] making vegetative reproduction of this species less efficient than generative reproduction. The flowers of *R. asiaticus* with over 30 stamens and circa 660 pistils are capable of producing almost 500 achenes for some cultivars and none for others. Therefore, the efficiency of generative reproduction in this species depends on the breeding system of the cultivar [52] and poses a challenge to horticultural production rather than species conservation. 

Our research has shown that *R. illyricus* can be successfully propagated under ex situ conditions and plant material can be obtained for active conservation treatments. The plants grown can be used depending on whether they are needed to enhance existing populations or to establish new ones for replacement populations. The vegetative propagation of the species is very efficient and produces a large number of progeny plants in a short time (each season, up to 3–4 progeny plants could be produced per mother plant). Although generative propagation is limited by low fruit-setting efficiency and low seed-germination capacity, it is also possible to obtain progeny plants in this way. *R. illyricus* is described as a self-incompatible species, so it is worth making such an effort to increase genetic diversity within the population and to facilitate cross-pollination between individuals representing different genotypes. Mature achenes can be collected from natural sites or a conservation collection, but it is doubtful that their direct sowing on natural sites can bring satisfactory results. Rather, we recommend that collected seeds should be further handled under ex situ conditions and stratified and/or germinated in low temperatures. The resulting seedlings and later adult plants and their progeny should be carefully labelled and cultivated as separate clones (progeny of one individual of generative origin). Such a procedure requires at least a few years; in our study, the first progeny clusters were formed in two-year-old plants obtained from seed, whereas all three-year-old plants were vegetatively propagating. The first plant obtained from seed also flowered only after three years. It is important to maintain an ex situ plant collection over a long time with the aim of gradually increasing genetic diversity (by obtaining seedlings), and at the same time some plants representing the different clones can be used to feed natural populations. Currently, we have 17 clones obtained from seed in our collection. Based on the observations of the seasonal cycle of *R. illyricus*, the optimal time for conservation measures involving the introduction of plants into the environment may be the dormancy period, which lasts about three months from July to September. It is recommended to plant dormant clusters of tubers in plots cleared of turf, a dozen or so at a time, mixing plants belonging to different clones.

The method of active conservation of *R. illyricus* proposed in the manuscript can be applied to different populations of this species, both in Poland and throughout its geographical range, where the species is losing natural resources. In our opinion, it may also be useful for various species requiring conservation and, in particular, to plants that pursue two modes of reproduction. Rare species having rapidly declining populations undergo a loss of genetic diversity that can have demographic consequences. Therefore, it is preferable to reintroduce plants using seedlings as it is most beneficial for enriching the genetic variability of the population being enhanced or restored [53,54]. However, this is not always easy due to the limitations of generative reproduction, e.g., lack of a partner for mating in self-incompatible plants, low pollination success, or low germination rates [4,55,56,57]. For example, in the clonal species *Lysimachia asperulifolia*, an effective population feeding treatment was carried out using rhizomes (ramets) [58]. In this case, it can be assumed that if plants were obtained from seed at an earlier stage and then propagated vegetatively, the use of such material would not only contribute to an increase in population size but also to n enhanced genetic diversity. However, each species and even population requires individual treatment and prior recognition of biology and threats [4].

## 5. Conclusions

This is the first time the annual developmental cycle of *R. illyricus* has been described, allowing us to present its ability to reproduce both generatively and vegetatively.The efficiency of vegetative propagation ex situ depended on the age of the tuber (clone) and indirectly on weather conditions. After three years, the best clones could produce up to 57 progeny clusters, which flowered in the first vegetative season, but the regeneration potential of the tubers started to decrease in the case of the two-year-old tubers.The high potential of *R. illyricus* for generative reproduction was limited by low seed-setting efficiency under ex situ conditions and difficulties with seed germination and seedling survival. In addition, the first flowering plant of seed origin was observed in the third year after planting.Vegetative reproduction was more effective than generative reproduction because more progeny clusters could be obtained during one season and they were able to propagate through both reproduction modes in the following season.The best way to increase the natural resources of this species would be ex situ generative propagation followed by vegetative propagation of the resulting plants.

## Figures and Tables

**Figure 1 biology-11-00873-f001:**
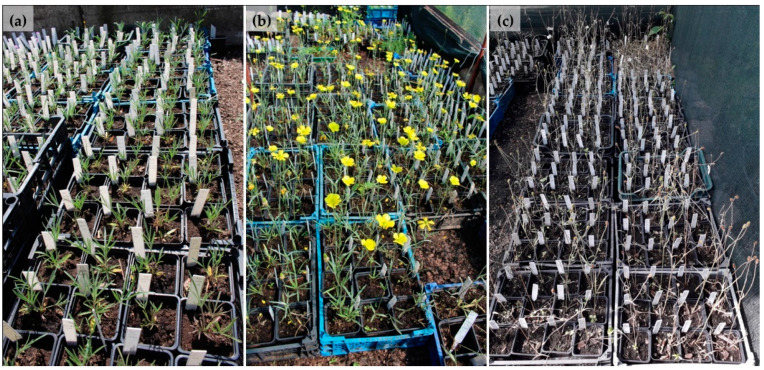
Labelled *Ranunculus illyricus* specimens growing in the collection: (**a**) vegetative aboveground shoots (20 April 2019); (**b**) flowering (28 May 2019); (**c**) ripening of fruit and die-back of aboveground shoots (24 June 2019).

**Figure 2 biology-11-00873-f002:**
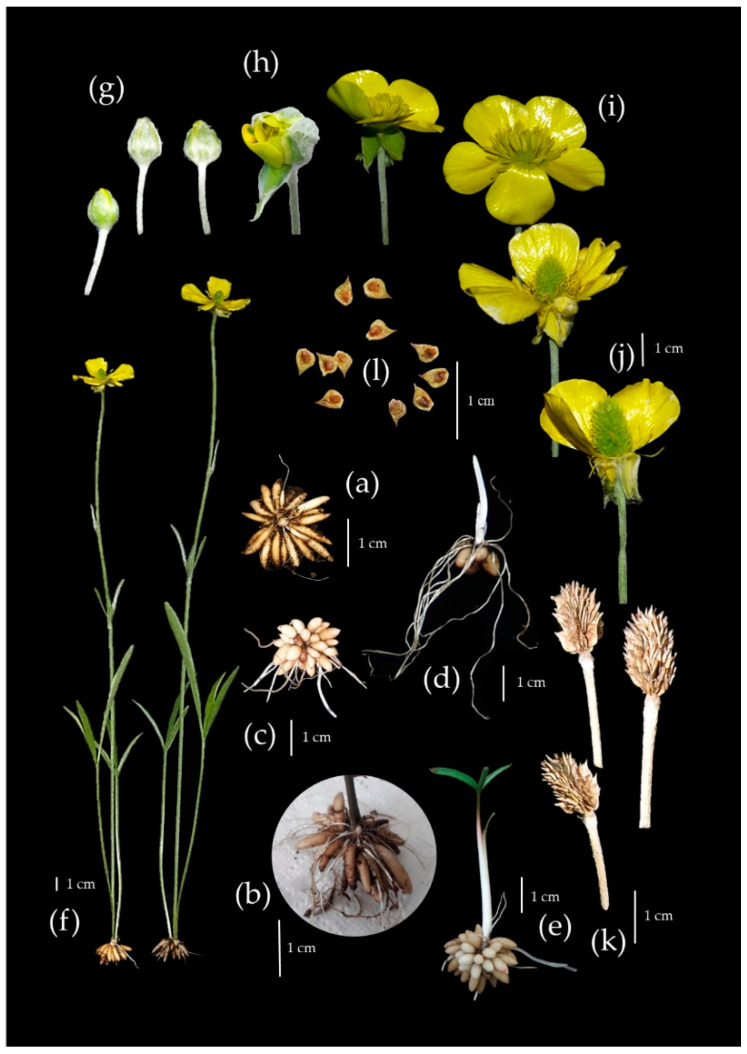
Annual developmental cycle of *Ranunculus illyricus*: (**a**) dormant cluster of tubers, (**b**–**d**) development of underground parts and (**e**) shoot with leaf after dormancy, (**f**) flowering plants, (**g**–**j**) developmental stages of flower, (**k**) receptacle with achenes, (**l**) ripe fruits.

**Figure 3 biology-11-00873-f003:**
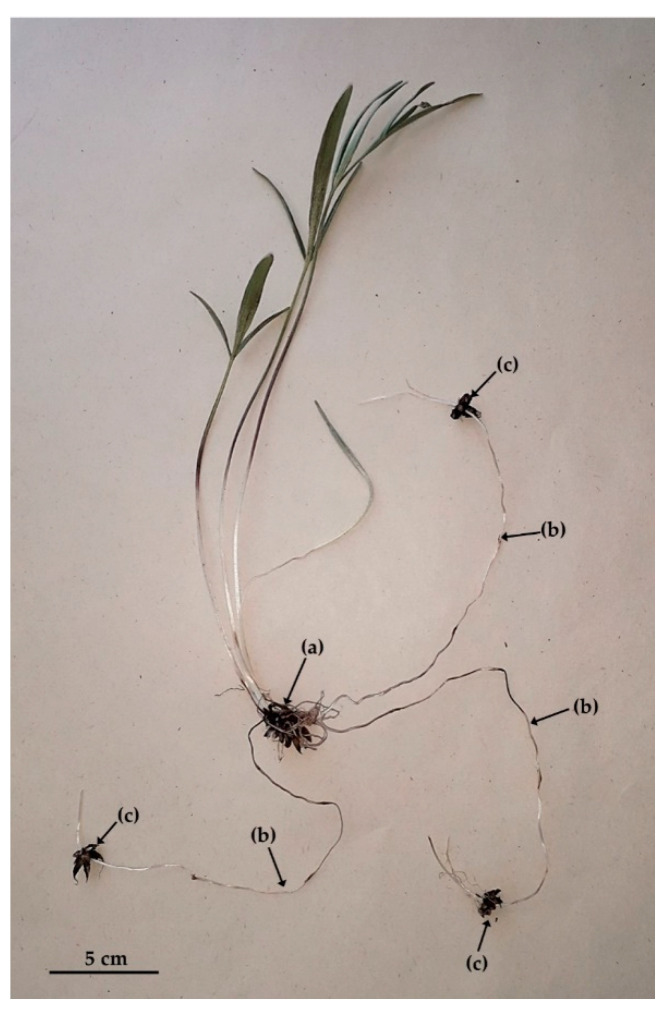
Vegetatively reproducing individual of *Ranunculus illyricus*: (**a**) mother cluster of tubers with aboveground shoot, (**b**) underground stolons, (**c**) progeny clusters of tubers.

**Figure 4 biology-11-00873-f004:**
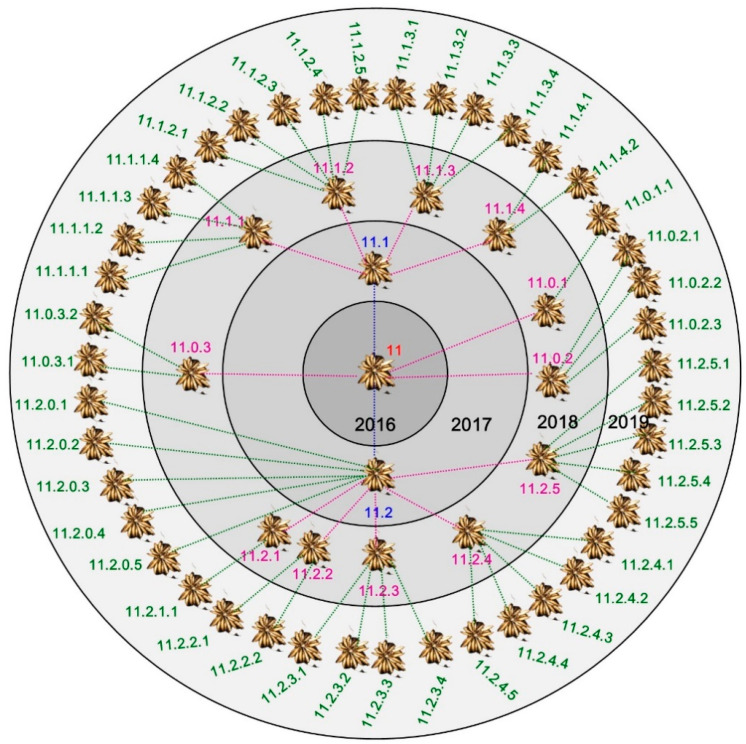
Vegetative development of one *Ranunculus illyricus* clone during a three-year period. Example: 11–mother cluster planted in 2016 produced two progeny clusters in 2017 (11.1 and 11.2) and three in 2018 (11.0.1, 11.0.2 and 11.0.3). In 2018, individual 11.1 produced four progeny clusters of tubers (11.1.1–11.1.4).

**Figure 5 biology-11-00873-f005:**
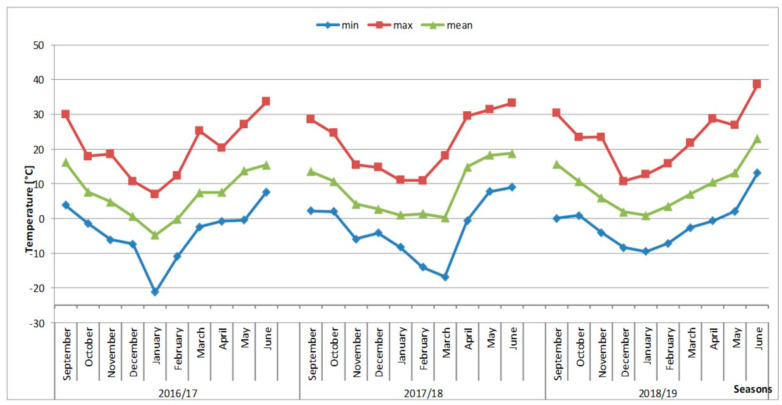
Daytime temperatures during the growing season of *R. illyricus* (September–June) in the years 2015–2019.

**Figure 6 biology-11-00873-f006:**
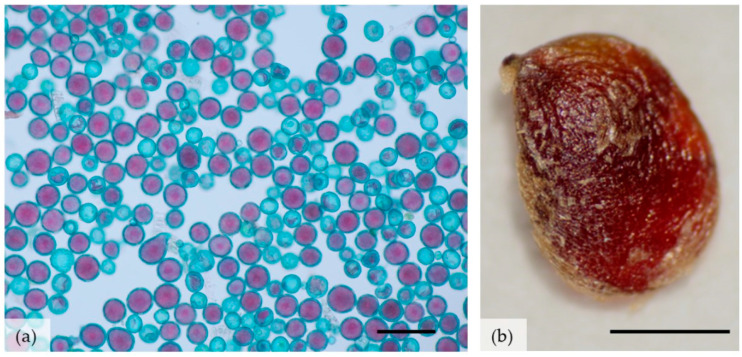
Viability of pollen and seeds of *Ranunculus illyricus*, (**a**) pollen after Alexander staining: red pollen grains are viable; (**b**) seed after tetrazolium staining—living tissue is red. Scale bars: (**a**) 100 μm, (**b**) 1 mm.

**Figure 7 biology-11-00873-f007:**
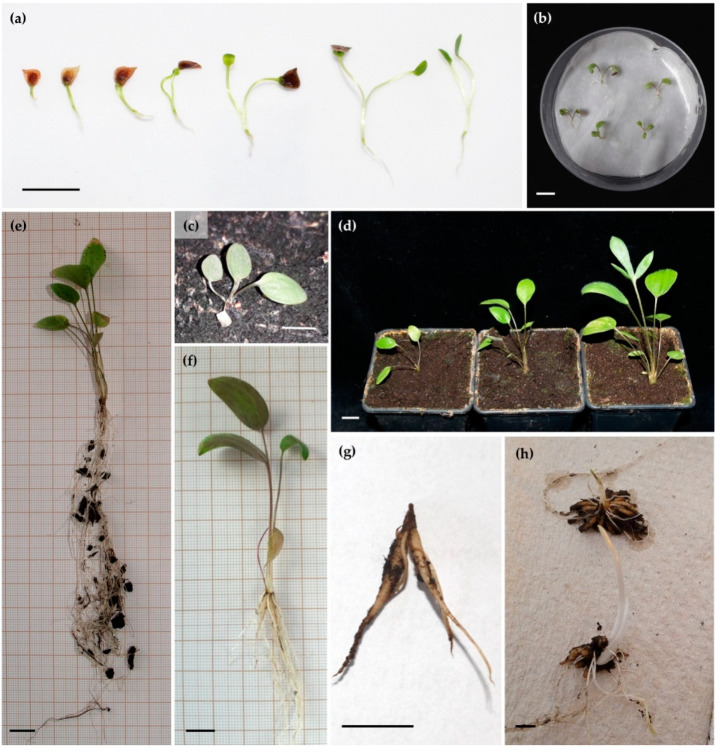
Seed germination and seed-derived plant development of *R. illyricus* under ex situ conditions: (**a**) epigeic seed germination between 10–20 days after sowing, (**b**) seedlings with cotyledons before potting, (**c**) two-month-old seedling with dried cotyledons and juvenile leaves, (**d**) three-month-old seed-origin plants, (**e**) morphological structure of above- and belowground organs of a seed-origin plant at the end of the first growing season, (**f**) individual of generative origin with some tuberous roots at the end of the first growing season, (**g**) cluster of tubers after the first overwintering, (**h**) three-year-old plant obtained from seed (top) and its vegetative progeny (bottom) before winter. Scale bar = 1 cm.

**Table 1 biology-11-00873-t001:** The fresh matter (FM) of tuber clusters, the number of tubers in one cluster, and number of progeny clusters (PC) for different age clusters in the years 2017–2019.

Age of Cluster	Number of Clusters	FM of Tuber Cluster ± SE [mg]	Number of Tubers in One Cluster ± SE	Number of PC * Produced by One MC **
New	377	573.5 ± 16.9 b ***	13.9 ± 0.21 a	Have not produced PC yet
One year	85	682.3 ± 43.0 b	22.1 ± 0.99 b	2.23 b
Two years	32	392.3 ± 46.5 a	16.5 ± 1.15 a	1.78 a
		Kruskal–Wallis and Dunn’s test	Kruskal–Wallis and Dunn’s test	Student *t*-test

* PC—progeny cluster, ** MC—mother cluster, *** a, b—values within a column followed by the same letter are not significantly different at *p* = 0.05.

**Table 2 biology-11-00873-t002:** Effect of tuber cluster age on the number of flowers produced.

Age of Cluster	Number of Clusters	Number of Flowers per Cluster ± SE
New	0	Have not flowered yet
One year	284	1.7 ± 0.08 b *
Two years	105	1.3 ± 0.13 a
		Mann–Whitney U test

* a, b—values within a column followed by the same letter are not significantly different at *p* = 0.05.

**Table 3 biology-11-00873-t003:** The size of tuber clusters and the reproduction mode for two age groups of clusters in the year 2018 and their effect on selected traits related to the efficiency of generative and vegetative propagation.

Mode of Reproduction in 2018	VG *	V	G	VG	V	G
**Assessed Parameter**	**Clusters**					
	**“New” 2017 (*n* = 64)**			**“One-Year” 2017 (*n* = 31)**		
Number of tubers in cluster in 2017	16.0 a **	15.0 a	15.5 a	26.9 a	24.1 a	26.2 a
Fresh matter of clusters in 2017 [mg]	1020 ab	700 a	1100 b	1740 a	1150 a	1010 a
	**“One-year” 2018**			**“Two-year” 2018**		
Number of flowers in 2018	2.2 a	×	1.5 a	1.7 a	×	1.4 a
Percentage of flowers setting fruits in 2018	79.1 a	×	50 a	56.1 a	×	50 a
Height of flowering stem in 2018 [cm]	30.4 a	×	19.9 a	25.7 a	×	25.2 a
Number of tubers in clusters in 2018	26.7 b	19.9 a	×	16.9 a	14.7 a	×
Fresh matter of clusters in 2018 [mg]	850 b	560 a	×	430 a	300 a	×

* VG—vegetatively and generatively, V—vegetatively, G—generatively ** a, b—values in rows for the same age followed by the same letter are not significantly different according to Kruskal–Wallis and Dunn’s test as well as Mann-Whitney U test and *p* = 0.05.

**Table 4 biology-11-00873-t004:** Correlation matrix of mean temperatures and size of tuber cluster (its FM and number of tubers).

	Mean Temperatures			
	March	April	May	June
Number of tubers in cluster	r = 0.2551 *p* = 0.000	r = −0.2560 *p* = 0.000	r = −0.2448 *p* = 0.000	r = 0.0016 *p* = 0.975
Fresh matter of cluster	r = 0.4120 *p* = 0.000	r = −0.4624 *p* = 0.000	r = −0.3812 *p* = 0.000	r = −0.1722 *p* = 0.000

**Table 5 biology-11-00873-t005:** The production and viability of pollen and efficiency of fruit setting of *Ranunculus illyricus*.

	Year of Evaluation	Mean ± SE	Test
Number of pollen grains in one anther [pcs]	2019	2106.3 ± 990.8	-
Number of stamens in one flower [pcs]	2019	66.3 ± 1.6	-
Number of pollen grains in one flower [pcs]	2019	139,644 ± 69,822	-
Viability of pollen grains [%]	2017	61.8 ± 1.2 ab *	ANOVA and Tukey test *p* = 0.05
	2018	68.5 ± 2.1 b	
	2019	53.6 ± 4.9 a	
Number of pistils in one flower [pcs]	2017	138 ± 7.6 a	ANOVA and Tukey test *p* = 0.213
	2018	154.7 ± 8.3 a	
	2019	148.0 ± 3.4 a	
Number of achenes in one flower [pcs]	2017	18.9 ± 3.0 b	Kruskal–Wallis and Dunn’s test *p* = 0.0000
	2018	6.0 ± 1.6 a	
	2019	13.3 ± 1.8 b	
Effectiveness of fruit set [%]	2017	12.8 ± 1.7 c	Kruskal–Wallis and Dunn’s test *p* = 0.0000
	2018	3.7 ± 0.9 a	
	2019	8.9 ± 1.1 b	

* a, b, c—values within a column for one assessed feature followed by the same letter do not differ significantly.

**Table 6 biology-11-00873-t006:** The effect of hot stratification and temperature on germination ability of *Ranunculus illyricus* seeds [% per plate ± SE].

Temperature of Germination	Hot Stratification		Means for Temperature of Germination
	Without Hot Stratification	Hot Stratification (50 °C)	
20 ± 2 °C	0.0 ± 0.0 a *	0.0 ± 0.0 a	0.0 ± 0.0 *A*
10 °C	40.0 ± 8.1 c	20.0 ± 8.1 b	30.0 ± 6.5 *B*
Means for stratification	20.0 ± 8.5 A	10.0 ± 1.8 A	

* a, b, c, A, *A*, *B*—values within columns and rows followed by the same letter do not differ significantly for *p* = 0.5 and Tukey test; two-way ANOVA was performed: one factor: hot stratification (upper case), second factor: temperature of germination (upper case italics), interaction: hot stratification × temperature of germination—lower case.

**Table 7 biology-11-00873-t007:** The effects of cold stratification and gibberellin application on germination ability of *Ranunculus illyricus* seeds [% per plate ± SE].

Application of GA_3_	Cold Stratification		Means for GA_3_ Application
	Without Stratification	Cold Stratification (4 °C)	
−GA_3_	0.0 ± 0.0 a *	15 ± 9.6 ab	7.5 ± 6.9 *A*
+GA_3_	5.0 ± 5 a	30 ± 12.9 b	17.5 ± 11.25 *A*
Means for stratification	2.5 ± 3.5 A	22.5 ± 11.3 B	

* a, b, A, B, *A*—values within columns and rows followed by the same letter do not differ significantly for *p* = 0.5 and Tukey test; two-way ANOVA was performed: one factor: cold stratification (upper case), second factor: GA_3_ application (upper case italics), interaction: cold stratification × GA_3_ application—lower case.

## Data Availability

The data presented in this study are available on request from the corresponding author.

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
