# Peer review of "Reproductive Biology of Dry Grassland Specialist *Ranunculus illyricus* L. and Its Implications for Conservation"

_biology, 2022, doi:10.3390/biology11060873_

Round 1
Reviewer 1 Report
I recommend to accept the MS with minor revisions according to the comments I have posted in the MS.Please see the attached file.

Author Response
Dear Reviewer,
We appreciate the time and effort that you dedicated to providing feedback on our manuscript “Reproductive biology of dry grassland specialist Ranunculus illyricus L. and its implications for conservation” and we are grateful for insightful comments and valuable improvements to it. We have carefully incorporated your’s suggestions. The changes are highlighted within the revised manuscript. Please see below a point-to-point response to the comments. All line numbers refer to the revised manuscript file with tracked changes.
We hope that you find our responses satisfactory.
Specific comments:
L 149-150 Staining method - Appropriate information has been added to Material and Methods chapter (in the revised manuscript lines 151-153)
L 203 ? - It has been corrected - “witch” changed to “with” (in the revised manuscript line 206)
L 263 ? - Text has been rewritten to make it simple (in the revised manuscript line 269)
L 275 please see comment in table 1 and correct it accordingly along the MS - We explain this below
L 276 - FM instead fresh matter has been introduced (in the revised manuscript line 285)
Table 1, column 2. - The age characteristic refers to tuber clusters and not to plants therefore we changed “objects” to “clusters”.
Table 1, column 4.: it is not clear what is meant by the use of tuber. Each cluster has several storage roots. But more than one tuber = cluster can be produced by one plant. Clarify the definitions along the MS”.
A cluster consists of tubers (a few to several dozen), which are thickened storage roots (please see lines 131-139, 218-223 in the revised manuscript). And yes that's right, the authors look at how many tubers go into one cluster.
Table 2, column 2,
“objects” has been changed into “cluster”
Table 6 and 7 – the average between temps of germination or stratification treats is meaningless (bottom row, right column). delete row. however- you can test for significance between the stratification treatments within each temp and between temps within each stratification treat.
The authors would prefer to stay with the current presentation of results.
The experiments were in a factorial design and the two-factor ANOVA showed significant differences between some factors:
Table 6. One factor “hot stratification”, the other factor “germination temperature”. It is hard to agree that germination temperature (regardless of using hot stratification or not) is meaningless since statistical analysis showed significant differences (right row, upper case italics). Which means that no matter how the seeds were treated they germinated better at lower temperature. The significance of the interaction of factors (between the stratification treatments within each temp and between temps within each stratification treat) is in the central part of the table – lower case letters. It was also shown that there is no significant difference between the combinations within the hot stratification factor (bottom row, upper case), and because the lack of a significant difference is also a result. We think it is better to include bottom row. However, to make easier to understand some explanation to the table body and footnotes have been added.
Table 7. The same arguments could be applied to the Table 7.
L 372 – „and” has been replaced „to” (in the revised manuscript line 392)
L 389 – In MS is written 10-18. - A germinating seed is considered to be one in which a radicle appears, and seed germination time was thus defined. Radicles appeared from 10th to 18th day after sowing (10-18). The photograph shows a seedling with cotyledons, hence the extra two days (10-20) (in the revised manuscript line 413)

Reviewer 2 Report
The manuscript "Reproductive biology of dry grassland specialist Ranunculus illyricus L. and its implications for conservation" is very good - it's based on well conducted experiment, it's clearly written and illustrated, and it has valuable practical recommendations for species conservation.
Another quality of the manuscript is that the authors acknowledged honestly some limits of their experiment - for example (L444-446): “In future it should be verified what the efficiency of generative reproduction of this species and what the proportion of individuals of generative versus vegetative origin would be in natural populations”. Nevertheless, this remark doesn't diminish the value of the study, since a low genetic diversity can occur in small in-situ populations, like that of Miernow site. It's even a better fit.
I have only minor/formal corrections or suggestions (the paper will be perfect without them, anyway).
Chapter 2.1 >> Beside soil conditions, please describe the “atmosphere” of the experiment. Were the pots deposited in open space or in greenhouse? Were the plants covered naturally by snow? Or, how were they protected during winter? If needed, use such information in the following chapters where you discuss the survival rates over winter.
L115: Explain what “selected clones” represent. Which were the reason and the selection criteria?
L127: “Each cluster was then replanted” >> replanted separately in new pots, right? It can be understood, but it’s better to say it.
L201: “was tested witch the Shapiro” >> see “witch”
Table 1 & 2, column 2: “Number of objects” >> Can you say "clusters" instead of "objects"?
Table 3, in the 6th column: Try to adjust the placement of “a” (on the right side of values, not below)
L303-304: “It turned out that FM of one-year tuber clusters did not determine the mode of reproduction in the following year. However, the FM of the younger clusters – “new” clusters – did.” >> I recommend to avoid “determine”. It could be merely a coincidence. From the statistical analysis alone (presented in table 3) you couldn’t establish the causal relationship. Can you? You’ll need repetitions of such observations. The causal factor could be related with hormones, some inhibitory processes, or else. You may choose some neutral words, like “correlate” or “co-vary”.
L334: leave an empty row between table and sub-chapter 3.3.
Table 5: add units for Viability of pollen grains >> [%] ?
L489: F. verna >> Use consistently R. ficaria.
L489-491: “F. verna was also found to exhibit low efficiency of generative reproduction [32], which does not prevent it from occurring in large numbers thanks to its efficient vegetative reproduction.” >> Indeed. This efficiency is probably due to lower inter-specific competition on the ground of deciduous forests (with R. ficaria), comparing with grasslands.
L491-493: “It is known that in undisturbed communities this mode of reproduction dominates and effectively ensures the survival of plants.” >> This sentence requires citation and few details on its validity (for all species? all habitats? on long term?). Vegetative reproduction may ensure survival, but not necessarily the optimal genetic structure, as authors already showed elsewhere.
L509-510: “The vegetative propagation of the species is very efficient and produces a large number of progeny plants in a short time” >> Can you recommend watering ? Or it isn’t the case?
L511: “up to 3.3 progeny plants” >> being something real, not statistic, you may say “up to 3-4 progeny plants”.
Author Response
Dear Reviewer,
We appreciate the time and effort that you dedicated to providing feedback on our manuscript “Reproductive biology of dry grassland specialist Ranunculus illyricus L. and its implications for conservation” and we are grateful for insightful comments and valuable improvements to it. We have carefully incorporated your’s suggestions. The changes are highlighted within the revised manuscript. Please see below a point-to-point response to the comments. All line numbers refer to the revised manuscript file with tracked changes. We are very pleased to meet with such a positive review of our manuscript - for which we thank you very much.
We hope that you find our responses satisfactory.
Specific comments:
Chapter 2.1 >> Beside soil conditions, please describe the “atmosphere” of the experiment. Were the pots deposited in open space or in greenhouse? Were the plants covered naturally by snow? Or, how were they protected during winter? If needed, use such information in the following chapters where you discuss the survival rates over winter.
Plants were cultivated in open field conditions, naturally covered by snow during winter (in the revised manuscript lines 111-112).
L115 - Explain what “selected clones” represent. Which were the reason and the selection criteria?
As the number of plants gradually increased with successive seasons in the last year of observation, we randomly selected some clones for observations. “Randomly” has been added to the manuscript (in the revised manuscript line 116).
L127 - “Each cluster was then replanted” >> replanted separately in new pots, right? It can be understood, but it’s better to say it.
„separately in new pots” has been added in the line 128.
L201 - “was tested witch the Shapiro” >> see “witch”
“Witch” has been changed into “with” (in the revised manuscript line 206).
Table 1 & 2, column 2: “Number of objects” >> Can you say "clusters" instead of "objects"?
It has been changed into “clusters”.
Table 3, in the 6th column: Try to adjust the placement of “a” (on the right side of values, not below)
The table has been edited again so that the "a" fits to the right of the value.
L303-304 - “It turned out that FM of one-year tuber clusters did not determine the mode of reproduction in the following year. However, the FM of the younger clusters – “new” clusters – did.” >> I recommend to avoid “determine”. It could be merely a coincidence. From the statistical analysis alone (presented in table 3) you couldn’t establish the causal relationship. Can you? You’ll need repetitions of such observations. The causal factor could be related with hormones, some inhibitory processes, or else. You may choose some neutral words, like “correlate” or “co-vary”.
“Determine” has been replaced by ”co-vary”, thanks for your suggestion (in the revised manuscript line 314).
L334 - leave an empty row between table and sub-chapter 3.3.
The empty line has been added between the table and sub-chapter (in the revised manuscript line 348).
Table 5: add units for Viability of pollen grains >> [%] ?
The units have been completed.
L489 - F. verna >> Use consistently R. ficaria.
In fact, the current name is Ficaria verna and R. ficaria is an older name, now synonymous. For this reason, we switched the order of the names in line 508 and left F. verna in the text as before.
L489-491 - “F. verna was also found to exhibit low efficiency of generative reproduction [32], which does not prevent it from occurring in large numbers thanks to its efficient vegetative reproduction.” >> Indeed. This efficiency is probably due to lower inter-specific competition on the ground of deciduous forests (with R. ficaria), comparing with grasslands.
Good point. There is less competition in the undergrowth compared to the grassland communities. But since F. verna is not the main object of observation and discussion, we would not like to develop the discussion in this direction.
L491-493 - “It is known that in undisturbed communities this mode of reproduction dominates and effectively ensures the survival of plants.” >> This sentence requires citation and few details on its validity (for all species? all habitats? on long term?). Vegetative reproduction may ensure survival, but not necessarily the optimal genetic structure, as authors already showed elsewhere.
We have supplemented the references in this section of the text by referring to deciduous forest (Eriksson, O. Seedling dynamics and life histories in clonal plants. Oikos 1989, 55,231–238; Pirożnikow, E. Life cycle of herbaceous plants in disturbed and undisturbed sites of oak-linden-hornbeam forest (Tilio-Carpinetum). Ekol. Pol. 1998, 46, 157–168) and steppe communities (Czarnecka, B. The dynamics of the population of a steppe perennial Senecio macrophyllus M. BIEB. during xerothermic grassland overgrowing. Acta Societatis Botanicorum Poloniae 2009, 78(3), 247-256) (in the revised manuscript lines 517-518).
L509-510 - “The vegetative propagation of the species is very efficient and produces a large number of progeny plants in a short time” >> Can you recommend watering? Or it isn’t the case?
One can recommend watering but we didn't really have a comparative experiment to evaluate the need for watering and we do not know to what extent this is a critical factor for this xerothermic species. It probably depends on the weather season, and we simply tried to keep “optimum moisture content’.
L511 - “up to 3.3 progeny plants” >> being something real, not statistic, you may say “up to 3-4 progeny plants”.
“3.3” has been changed to 3-4 (in the revised manuscript line 536).

Reviewer 3 Report
This is a traditional common garden of plant biology. Although a lot of interesting results were found, it seems that the manuscript is needed to be improved. The discussion is limited to Ranunculus illyricus. It would be better to broad it about the endangered mechanism and population dynamics of other similar entities. In addition, it would be better to propose conservation plan for remaining populations in Poland.
Author Response
Dear Reviewer,
We are very grateful for your profound review and we also appreciate your suggestions, which have been very helpful in improving our manuscript “Reproductive biology of dry grassland specialist Ranunculus illyricus L. and its implications for conservation” and we are grateful for insightful comments and valuable improvements to it. We have carefully incorporated your’s suggestions:
“The discussion is limited to Ranunculus illyricus. It would be better to broad it about the endangered mechanism and population dynamics of other similar entities”
At the end of the discussion section, we have included a paragraph that addresses reproductive constraints in populations of endangered plants more generally, and we have provided an example of a species in which we believe the method proposed in the manuscript could be beneficial (please see lines 571 - 584 in the revised manuscript).
“In addition, it would be better to propose conservation plan for remaining populations in Poland.”
We hope that the method of conservation of the R. illyricus proposed in the manuscript can be applied to different populations of this species, both in Poland and throughout its geographic range, where the species is losing natural resources. We have supplemented this information in the discussion chapter (please see line 569-571 in the revised manuscript).
The changes are highlighted within the revised manuscript. All line numbers refer to the revised manuscript file with tracked changes.
We hope that you find our responses satisfactory.

Round 2
Reviewer 3 Report
The revision has answered all my concerns. Generally, the experiment is normal but is well designed. Althougth this is a traditional species biology of Ranunculus illyricus, authors have tried their best to improve the manuscript. The endangered mechanisms of Ranunculus illyricus and conservation strategy have been well illustrated.